# Phenotyping Fatigue Profiles in Marfan Syndrome Through Cluster Analysis: A Cross-Sectional Study of Psychosocial and Clinical Correlates

**DOI:** 10.3390/jcm14165802

**Published:** 2025-08-16

**Authors:** Nathasha Samali Udugampolage, Jacopo Taurino, Alessandro Pini, Edward Callus, Arianna Magon, Gianluca Conte, Giada De Angeli, Miriam Angolani, Giulia Paglione, Irene Baroni, Pasquale Iozzo, Rosario Caruso

**Affiliations:** 1Department of Biomedicine and Prevention, University of Rome “Tor Vergata”, 00133 Rome, Italy; giulia.paglione@grupposandonato.it (G.P.); pasquale.iozzo@policlinico.pa.it (P.I.); 2Cardiovascular Genetic Centre, IRCCS Policlinico San Donato, 20097 San Donato Milanese, Italy; jacopo.taurino@grupposandonato.it (J.T.); alessandro.pini@grupposandonato.it (A.P.); miriam.angolani@grupposandonato.it (M.A.); 3Clinical Psychology Service, IRCCS Policlinico San Donato, 20097 San Donato Milanese, Italy; edward.callus@unimi.it; 4Department of Biomedical Sciences for Health, University of Milan, 20133 Milano, Italy; 5Health Professions Research and Development Unit, IRCCS Policlinico San Donato, 20097 San Donato Milanese, Italy; arianna.magon@grupposandonato.it (A.M.); gianluca.conte@grupposandonato.it (G.C.); 6Clinical Research Service, IRCCS Policlinico San Donato, 20097 San Donato Milanese, Italy; giada.deangeli@grupposandonato.it (G.D.A.); irene.baroni@grupposandonato.it (I.B.)

**Keywords:** Marfan syndrome, fatigue, cluster analysis, t-SNE, psychosocial profiles, precision medicine

## Abstract

**Background/Objectives**: Fatigue is a highly prevalent and burdensome symptom among individuals with Marfan syndrome (MFS), yet its heterogeneity and underlying psychosocial and clinical correlates remain underexplored. This study aimed to identify and characterize distinct fatigue-related profiles in MFS patients using a data-driven cluster analysis approach. **Methods**: A cross-sectional study was conducted involving 127 patients with MFS from a specialized connective tissue disorder center in Italy. Participants completed self-reported measures of fatigue severity (Fatigue Severity Scale, FSS), depressive symptoms (Patient Health Questionnaire-9, PHQ-9), and insomnia (Insomnia Severity Index, ISI). The body mass index (BMI) and clinical data were also collected. A t-distributed stochastic neighbor embedding (t-SNE) analysis was performed to reduce dimensionality, followed by hierarchical clustering (Ward’s method), exploring solutions from k = 2 to k = 10. The optimal cluster solution was identified based on silhouette scores and clinical interpretability. **Results**: Three distinct clusters emerged: (1) a cluster characterized by low fatigue with minimal psychological and sleep-related symptoms (younger patients, lower PHQ-9 and ISI scores), (2) a cluster characterized by moderate fatigue with moderate psychological and sleep-related symptoms (intermediate age, moderate PHQ-9 and ISI scores), and (3) a cluster characterized by high fatigue with elevated psychological and sleep-related symptoms (older patients, higher PHQ-9, ISI, and FSS scores). Significant differences were observed across clusters in age, BMI, depressive symptoms, insomnia severity, and fatigue levels (all *p* < 0.05). **Conclusions**: Our findings highlight the heterogeneity of fatigue experiences in MFS and suggest the importance of profiling patients to guide personalized interventions. This approach may inform precision medicine strategies and enhance the quality of life for individuals with this rare disease.

## 1. Introduction

Marfan syndrome (MFS) is a rare, heritable connective tissue disorder characterized by the systemic involvement of the cardiovascular, musculoskeletal, and ocular systems [1,2,3,4]. Advances in medical and surgical management, particularly in cardiovascular care, have substantially improved survival rates and life expectancy for individuals with MFS [5,6,7]. However, despite these clinical advancements, many patients continue to report reduced quality of life, largely due to the burden of chronic symptoms that extend beyond life-threatening complications [8,9,10,11].

Individuals with MFS commonly experience a wide spectrum of symptoms that affect daily functioning and overall well-being [1,2,12,13]. Musculoskeletal manifestations such as joint hypermobility, scoliosis, pectus deformities, and chronic pain are prevalent and often contribute to physical limitations and activity restrictions [14]. Ocular complications, including lens dislocation and myopia, can impair vision and further limit daily activities [15]. Additionally, patients may encounter respiratory difficulties, gastrointestinal issues, and neurological symptoms, each adding to the cumulative burden of the condition [1,2,3,16,17,18]. While cardiovascular complications remain the primary focus due to their life-threatening nature, these multisystem symptoms collectively impact patients’ independence, social participation, and psychological health [19].

Among the various symptoms experienced by individuals with MFS, fatigue stands out as a highly prevalent and burdensome yet often under-recognized issue [8]. Recent studies have reported that up to 88% of adults with MFS experience significant fatigue, highlighting its widespread impact within this population [8,20,21,22,23]. Fatigue in MFS is not merely a feeling of tiredness but represents a profound and persistent sense of physical and mental exhaustion that interferes with daily activities. It has been associated with reduced work ability, decreased physical activity levels, impaired social participation, and an overall diminished quality of life [16,17]. Despite these far-reaching effects, fatigue is frequently overshadowed by the focus on life-threatening cardiovascular complications, leaving it under-addressed in clinical practice [13,14]. Nonetheless, patients consistently describe fatigue as one of the most limiting and distressing symptoms they face, underscoring the urgent need for better recognition and targeted management strategies [24].

Fatigue in MFS is increasingly recognized as a complex, multidimensional phenomenon that extends beyond purely physical limitations [21,22]. While connective tissue abnormalities and musculoskeletal impairments contribute to physical fatigue, psychological factors, such as depressive symptoms, anxiety, and reduced self-efficacy, are hypothesized to play a substantial role in modulating fatigue severity and perception [8]. Sleep disturbances, including insomnia and poor sleep quality, have also been identified as key contributors, potentially exacerbating fatigue and further diminishing daily functioning [8]. Moreover, clinical variables such as disease duration, body mass index, and comorbidities may interact with psychological and lifestyle factors, creating diverse fatigue experiences among individuals with MFS [9,25]. Importantly, this variability underscores the heterogeneity of fatigue in this population; not all patients are affected equally, nor do they share the same underlying contributing factors. Identifying these potential correlates is essential to uncovering distinct fatigue phenotypes and informing personalized management strategies [8,9,19].

Despite the growing recognition of fatigue as a significant burden in MFS, current research has primarily focused on describing its prevalence and exploring individual associations with clinical or psychosocial factors [8,26,27]. While these studies have provided valuable insights into potential contributors, they often rely on average group-level analyses that overlook important inter-individual differences. Consequently, patients with MFS are frequently viewed as a homogeneous group, masking the presence of distinct subgroups with unique fatigue profiles and underlying correlates [8].

To date, no studies have systematically classified individuals with MFS into meaningful subgroups based on an integrated set of psychosocial and clinical variables, such as depressive symptoms, insomnia severity, and other disease-related factors [8,9,19]. There is a critical need for approaches that move beyond simple comparisons and instead identify clinically relevant phenotypes, which could guide more personalized assessment and management strategies. For these reasons, this study aims to identify and characterize distinct fatigue phenotypes in individuals with MFS using a data-driven cluster analysis approach, integrating fatigue severity, depressive symptoms, insomnia, and clinical variables to uncover psychosocial and clinical profiles to inform tailored assessment and management strategies. In addition, based on prior evidence [20,28,29,30,31,32,33], we hypothesized the following: (a) Fatigue in MFS is a heterogeneous construct, with individuals clustering into discrete phenotypes differing in severity and psychosocial burden [20,31]. (b) Higher fatigue severity would be associated with greater depressive symptoms, more severe insomnia, and a higher BMI [28,29,32]. (c) Demographic and clinical characteristics, such as age and disease duration, would further distinguish these phenotypes [30,31,32].

## 2. Materials and Methods

### 2.1. Design

This study employed a cross-sectional design conducted at a single center, specifically a reference center for MFS in Italy [3]. The study was designed and reported in accordance with the Strengthening the Reporting of Observational Studies in Epidemiology (STROBE) guidelines to ensure transparency and methodological rigor [34].

### 2.2. Sample Size

A Monte Carlo simulation analysis was performed to explore the relationship between sample size and silhouette score, a key indicator of cluster separation quality [35,36]. We simulated cluster solutions ranging from k = 2 to k = 10, varying the sample size from 50 to 250 individuals in increments of 25. For each combination of sample size and k, 1000 iterations were conducted to ensure robust estimates of the silhouette score distribution. The simulation was performed under the hypothesis of moderately separated clusters (Cohen’s d = 1), reflecting a realistic yet meaningful differentiation between potential phenotypes [8].

The results (Figure 1) demonstrated that a sample size of approximately 125 patients represented the optimal balance, achieving adequate silhouette scores and ensuring stable and interpretable clustering, particularly for solutions with k < 5. Beyond 125 patients, silhouette improvements plateaued, indicating diminishing returns with further increases in sample size. This sample size was therefore chosen as it offers sufficient power to identify distinct fatigue-related phenotypes while maintaining practical feasibility in the context of a rare condition such as MFS.

### 2.3. Setting and Eligibility Criteria

This study was conducted at a single specialized center for heritable connective tissue disorders in Italy, which adopts a multidisciplinary approach integrating cardiologists, clinical geneticists, nurse practitioners, and psychologists [19]. This center is dedicated to providing comprehensive clinical care for patients with MFS, facilitating regular follow-ups in an outpatient setting. Eligible participants were identified from clinic records and invited to participate during routine follow-up visits or contacted by phone or email. To achieve the required sample size for optimal silhouette scores (n = 125), all eligible patients at the center (n = 170) were invited, as a response rate of approximately 75% was anticipated based on prior experience [8,9,16,18,37]. Ultimately, 127 patients agreed to participate, in line with the anticipated response rate.

Inclusion criteria were age ≥ 18 years, a confirmed diagnosis of MFS, and proficient Italian language skills (speaking, reading, and writing). MFS diagnoses were established according to the 2010 revised Ghent nosology, incorporating major criteria such as aortic root dilation, ectopia lentis, specific systemic features, and, when available, confirmation of the FBN1 gene mutation [4]. Patients with cognitive impairments were excluded to enhance internal validity as such impairments could confound self-reported measures of fatigue and related psychosocial variables. Cognitive impairment status was determined based on routine assessments documented in medical records within the last two years.

### 2.4. Procedure

In this study, 127 individuals agreed and completed the data collection process, resulting in a response rate consistent with expectations. Participants were asked to complete a series of self-report questionnaires administered through a secure web-based platform [38], which was specifically designed to ensure compliance with the General Data Protection Regulation (GDPR) and guarantee confidentiality and data privacy [39].

To ensure data accuracy and reliability, a validation process was implemented for self-reported clinical and anamnestic information. This included a cross-check by a clinician who had access to each patient’s electronic medical records, allowing the verification of self-reported data against documented medical history and clinical assessments. This approach strengthened the overall validity and robustness of the collected dataset.

### 2.5. Measures

Sociodemographic and clinical data were collected, including sex (male, female, or other), age (in years), educational level (primary school, lower secondary school, higher secondary school, or university), and the number of years since diagnosis. The body mass index (BMI) was calculated as weight in kilograms divided by height in meters squared (kg/m^2^).

Medical status was documented with a focus on cardiovascular involvement, including aortic root dilatation, aortic aneurysm, aortic dissection, and valvular pathologies, as well as other comorbidities such as scoliosis and visual impairments. Information on ongoing pharmacological treatments (e.g., beta-blockers, angiotensin receptor blockers) and mental health treatments (e.g., insomnia medications) was also collected.

Fatigue severity was assessed using the Fatigue Severity Scale (FSS), a validated self-administered instrument designed to evaluate the impact of fatigue on daily functioning [40]. The FSS consists of nine statements rated on a 7-point Likert scale (1 = strongly disagree to 7 = strongly agree). The final score is calculated as the mean of the item scores, with higher scores indicating more severe fatigue. A score of ≥4.67 suggests clinically significant fatigue, with a reported sensitivity of 82% and specificity of 87% for identifying disabling fatigue [40].

Depressive symptoms were measured using the Italian version of the Patient Health Questionnaire-9 (PHQ-9) [41], which assesses the presence and severity of depressive symptoms over the previous two weeks. Each of the nine items is scored from 0 (not at all) to 3 (nearly every day), resulting in a total score ranging from 0 to 27. Scores are interpreted as follows: 0–4 = no depressive symptoms, 5–9 = minimal depression, 10–14 = mild depression, 15–19 = moderate depression, and ≥20 = severe depression. Higher scores reflect greater symptom severity [41].

Insomnia severity was evaluated using the Italian version of the Insomnia Severity Index (ISI) [42], a 7-item self-report scale assessing the severity and impact of insomnia symptoms. Each item is rated on a 5-point Likert scale (0–4), with total scores ranging from 0 to 28. Scores are interpreted as follows: 0–7 = no clinically significant insomnia, 8–14 = subthreshold insomnia, 15–21 = moderate clinical insomnia, and 22–28 = severe clinical insomnia. Higher scores indicate greater insomnia severity [42].

### 2.6. Data Analysis

Data distributions were preliminarily inspected to verify completeness, identify outliers, and detect potential data entry errors [43]. No missing data were reported, as all self-reported measures were collected through mandatory fields in the online platform, ensuring complete datasets.

To reduce dimensionality and enhance the visualization of patient-level variability, t-distributed stochastic neighbor embedding (t-SNE) was performed [44]. Age, the BMI, and the fatigue severity score were used for t-SNE. The t-SNE analysis was conducted with two output dimensions (dims = 2), a perplexity value of 20 (balancing local and global structure), and a maximum of 500 iterations to ensure convergence. Perplexity is a key parameter in t-SNE, interpreted as a smooth measure of the effective number of nearest neighbors, which helps balance the focus between preserving local and global data structure in the resulting low-dimensional embedding. An initial principal component analysis (PCA) step was applied as part of the standard t-SNE preprocessing pipeline to improve computational efficiency and avoid local minima [45]. The resulting two t-SNE components effectively standardized and integrated information from variables with different measurement scales, enabling subsequent clustering analyses in a lower-dimensional space that preserves neighborhood relationships.

Once the two-dimensional t-SNE coordinates were obtained, hierarchical clustering using Ward’s minimum variance method was applied [46]. Solutions ranging from k = 2 to k = 10 clusters were explored. Cluster validity was evaluated through the average silhouette score, which quantifies cohesion and separation between clusters, and by assessing clinical interpretability. The optimal cluster solution was identified based on a combination of silhouette score trends, dendrogram inspection, and the clinical relevance of patient profiles [46,47]. After selecting the most informative solution, clusters were visualized in the t-SNE scatterplots, and dendrogram cuts were assessed to confirm the appropriateness of the partitioning.

Subsequently, inferential comparisons between clusters were performed to describe and characterize each group. For continuous variables (e.g., age, BMI, years since diagnosis), normality was preliminarily assessed. In cases of non-normal distributions, non-parametric Kruskal–Wallis tests were used, while ANOVA was applied when the assumptions were met. For categorical variables (e.g., sex, education level, depressive and insomnia categories), Pearson’s chi-square tests were employed to test for differences across clusters. To further explore the reviewer’s suggestion regarding possible linear or monotonic relationships between age, symptom burden, and fatigue, exploratory bivariate associations among age, years since diagnosis, BMI, fatigue severity (FSS), depressive symptoms (PHQ-9), and insomnia severity (ISI) were examined using Spearman’s rank correlation coefficients with Benjamini–Hochberg’s false discovery rate correction (α = 0.05) [48]. In addition, partial Spearman correlations were computed to assess the associations of fatigue with depressive symptoms and insomnia while controlling for age [49]. Finally, to investigate variability within older individuals, participants in the highest age tertile were subdivided into low- and high-fatigue groups based on the FSS clinical cut-off (4.67) and compared on psychosocial and clinical variables using the Mann–Whitney U test.

All statistical analyses were conducted in R (R Core Team, 2024. R: A Language and Environment for Statistical Computing, Version 4.5.0), using a two-tailed significance level set at α = 0.05.

## 3. Results

### 3.1. Sample Characteristics

A total of 127 patients with MFS were included in the analysis (see Table 1). The mean age was 38.8 years (SD ± 14.1), and 52% of the participants were female. Most patients had completed higher secondary school (37.8%) or university education (37.0%), and the median time since diagnosis was 13 years (IQR, 7–23 years). Regarding professional status, the majority were active workers in an office setting (55.9%), while 33.1% were unemployed. The median BMI was 21.3 kg/m^2^ (IQR: 19.1–23.4). Cardiovascular comorbidities were reported in 80.3% of participants, with 76.4% taking cardiovascular medications and 8.7% reporting hypertension. Other frequent comorbid conditions included scoliosis (68.3%), visual impairments (44.1%), and thyroid dysfunction (11.5%). The mean PHQ-9 score was 6.0 (SD ± 4.5), indicating generally mild levels of depressive symptoms. Specifically, 40.9% of patients reported no depressive symptoms, while 42.5% experienced minimal depressive symptoms. The mean ISI score was 7.0 (SD ± 4.1), with 61.4% of patients reporting no clinically significant insomnia. Fatigue severity, assessed via the FSS, had a mean score of 3.9 (SD ± 1.7), with 31.5% of participants meeting the threshold for clinically relevant fatigue (FSS score ≥ 4.67). Over the previous two weeks, 67.2% of patients reported experiencing fatigue occasionally, and 25.2% reported experiencing fatigue daily. In terms of fatigue pattern, the majority (65.5%) described fatigue mainly in the afternoon, 21.0% in the morning, and 13.4% throughout the entire day.

### 3.2. t-SNE

The two-dimensional t-SNE visualization (Figure 2) provides a projection of the sample based on age, BMI, and fatigue severity (FSS score), enabling the exploration of latent data structures prior to clustering. This approach enables the optimization of input information into standardized, low-dimensional variables that serve as carriers of relevant clinical and psychosocial information for the clustering procedure. In this scatterplot, individual points are colored by sex (blue for male and red for female) and sized according to FSS scores, with larger points indicating higher fatigue severity. Ellipses were overlaid to represent the distribution of males and females, highlighting the overall spread of each sex within the t-SNE space. Additionally, the wide variation in point sizes across both ellipses highlights substantial within-group heterogeneity in fatigue severity, underscoring the importance of identifying subgroups with distinct fatigue-related phenotypes, which could be determined through cluster analysis.

### 3.3. Hierarchical Clustering

Hierarchical clustering was performed using the Ward method on the t-SNE components to identify distinct fatigue-related phenotypes among individuals with MFS. Figure 3 displays the distribution of participants projected in the two-dimensional t-SNE space, with color coding indicating cluster membership for solutions ranging from k = 2 to k = 10.

Figure 4 presents the corresponding dendrograms for each k, along with the calculated silhouette widths as an indicator of cluster separation quality. The silhouette width values progressively decreased from k = 2 (0.622) to k = 10 (0.471), indicating that more granular solutions offered slightly reduced internal cohesion. After examining all solutions, k = 3 was ultimately selected as the most appropriate choice. This solution represented a balance between acceptable silhouette width (0.543), sufficient granularity to capture inter-individual variability, and clinical interpretability. The chosen k = 3 solution enabled the identification of three clinically meaningful fatigue profiles, facilitating the subsequent analysis of psychosocial and clinical correlates. This approach supported a nuanced understanding of fatigue heterogeneity in MFS, aligning with the study’s aim to inform tailored assessment and management strategies.

#### Optimal Cluster Solution

After examining all solutions, the three-cluster (k = 3) configuration was identified as the most suitable, as it balanced silhouette width and clinical interpretability. Table 2 summarizes the detailed characteristics and statistical comparisons among clusters.

Cluster 1 (N = 49) included the youngest participants, with a mean age significantly lower than the other clusters (22.6 ± 3.1 years, *p* < 0.001). This group also exhibited the lowest median BMI (20.0, IQR 18.4–21.4, *p* < 0.001) and the shortest disease duration (median 8.6 years, IQR 6.6–10.0, *p* = 0.0027). PHQ-9 scores were significantly lower (mean 3.5 ± 2.6, *p* = 0.016), indicating fewer depressive symptoms, and ISI scores were also the lowest (mean 4.3 ± 3.7, *p* = 0.029), reflecting minimal insomnia. Fatigue severity was the lowest among clusters, as reflected in the FSS score (mean 2.7 ± 0.8, *p* < 0.001), and no individuals in this cluster presented with clinically relevant fatigue (*p* < 0.001).

Cluster 2 (N = 32) showed an intermediate clinical profile, with a mean age of 36.4 ± 5.9 years and a median BMI of 21.8 (IQR 19.5–23.6). Disease duration was moderate (median 12.1 years, IQR 9.7–15.0). Participants had intermediate levels of depressive symptoms (PHQ-9 mean 5.8 ± 3.7) and insomnia severity (ISI mean 6.7 ± 4.5). Fatigue severity (FSS mean 3.5 ± 1.1) and the proportion of individuals with clinically relevant fatigue (25.1%) were also intermediate, demonstrating statistically significant differences compared to both Cluster 1 and Cluster 3 (all *p*-values < 0.05 where indicated).

Cluster 3 (N = 46) comprised the oldest participants (mean age 53.5 ± 6.1 years, *p* < 0.001), with the highest BMI (median 23.9, IQR 22.5–25.6, *p* < 0.001) and the longest disease duration (median 15.6 years, IQR 13.7–19.0, *p* = 0.0027). This group reported the highest levels of depressive symptoms (PHQ-9 mean 7.2 ± 3.6, *p* = 0.016) and the most severe insomnia (ISI mean 8.8 ± 4.8, *p* = 0.029). Fatigue severity was markedly higher (FSS mean 4.9 ± 1.0, *p* < 0.001), and two-thirds of participants (67.4%) exhibited clinically relevant fatigue (*p* < 0.001), indicating a substantial psychosocial and clinical burden. No statistically significant differences were observed in sex distribution across clusters (*p* = 0.137), suggesting that sex did not contribute to cluster differentiation in this cohort.

Based on the integrated analysis of fatigue severity, depressive symptoms, insomnia, and clinical variables, the three clusters were labeled to reflect their empirical characteristics. Cluster 1, comprising younger patients with the lowest fatigue scores, minimal depressive symptoms, and lower levels of insomnia, was designated as the “low fatigue with minimal psychological and sleep-related symptoms” profile. Cluster 2 represented an intermediate group, characterized by moderate levels of fatigue, depressive symptoms, and insomnia, and was labeled as the “moderate fatigue with moderate psychological and sleep-related symptoms” profile. Cluster 3 included older patients exhibiting the highest levels of fatigue, depression, and insomnia, along with higher BMI values, and was labeled as the “high fatigue with elevated psychological and sleep-related symptoms” profile. This naming convention aims to reflect the observed data patterns without inferring unmeasured psychological constructs.

To further illustrate the separation and internal coherence of the identified fatigue-related phenotypes, we plotted the three clusters using a two-dimensional t-SNE scatter plot (Figure 5). Each point represents an individual patient, with color indicating cluster membership and point size reflecting the individual FSS score. Ellipses were added to capture the approximate 95% confidence region for each cluster, emphasizing the compactness and distinctiveness of the groups in the reduced-dimensionality space. This graphical representation confirms that the clusters maintain clear separation and limited overlap, supporting both their statistical validity and clinical interpretability. The t-SNE visualization thus provides an intuitive depiction of how the clusters diverge with respect to age, the BMI, and fatigue severity, strengthening confidence in the identified fatigue-related profiles.

### 3.4. Exploratory Associations Between Age, Psychosocial Factors, and Fatigue

Age showed a moderate positive correlation with the BMI (ρ = 0.51, *p* < 0.001) and weaker but statistically significant associations with fatigue severity (ρ = 0.22, *p* = 0.014) and insomnia severity (ρ = 0.18, *p* = 0.042). Fatigue severity correlated strongly with depressive symptoms (ρ = 0.62, *p* < 0.001) and moderately with insomnia severity (ρ = 0.41, *p* < 0.001), but not with the BMI (ρ = 0.04, *p* = 0.634).

Partial correlation analyses controlling for age confirmed that fatigue severity remained strongly associated with depressive symptoms (ρ = 0.60, *p* < 0.001) and moderately with insomnia severity (ρ = 0.39, *p* < 0.001), indicating that these relationships are largely independent of age.

To further explore variability within older participants, those in the highest age tertile (≥47 years) were stratified into low-fatigue (FSS < 4.67) and high-fatigue (FSS ≥ 4.67) groups. Within this subgroup, high-fatigue individuals reported significantly higher PHQ-9 scores (9.53 ± 4.56 vs. 5.00 ± 3.76, U = 90.5, *p* = 0.0008) and a trend toward higher ISI scores (9.00 ± 4.24 vs. 6.67 ± 3.03, U = 158.5, *p* = 0.087), whereas the BMI and years since diagnosis did not differ meaningfully (both *p* ≥ 0.66).

## 4. Discussion

This study aimed to identify distinct fatigue-related phenotypes among patients with MFS, leveraging advanced clustering approaches to capture the heterogeneity of clinical presentations. Through the integration of t-SNE for dimensionality reduction and hierarchical clustering, we successfully delineated three clinically meaningful clusters: one characterized by low fatigue with minimal psychological and sleep-related symptoms, a second with moderate fatigue and intermediate levels of depressive and insomnia symptoms, and a third with high fatigue and elevated psychological and sleep-related symptoms.

Fatigue is increasingly recognized as a significant and debilitating symptom in hereditary connective tissue disorders such as MFS, profoundly impacting patients’ functional status and quality of life [8,15,22,25,27]. Our findings highlight that fatigue in MFS is not a uniform experience, but rather a complex and multifaceted phenomenon influenced by demographic, psychological, and clinical factors [44,50]. Importantly, by applying a data-driven clustering strategy supported by t-SNE, we were able to generate low-dimensional representations of patients’ characteristics, thus optimizing the interpretability of the clusters while maintaining the richness of the underlying data [51]. This methodological approach enabled the identification of homogeneous subgroups with clear clinical profiles, offering new insights into the variability of fatigue experiences in MFS and paving the way for more personalized care strategies.

Our findings are consistent with the previous literature highlighting fatigue as a highly prevalent and impactful symptom among individuals with MFS [8,24]. Prior studies have reported fatigue prevalence rates ranging from 30% to over 80% in patients with MFS, often emphasizing its significant detrimental effect on daily functioning and health-related quality of life [8,20,22,52]. Moreover, fatigue has been shown in previous published research to correlate with psychological distress, particularly depression and anxiety, and to be exacerbated by sleep disturbances such as insomnia [8,24]. The identification of three distinct clusters in our analysis reinforces the notion of fatigue as a multifactorial phenomenon in MFS, supporting existing hypotheses that fatigue severity is shaped not only by physical factors but also by psychological and lifestyle-related components. A notable novelty of our work is the clear emergence of a cluster characterized by low fatigue and minimal psychological and sleep-related symptoms. While previous studies have suggested the existence of individuals with better fatigue adaptation, this specific subgroup has not been explicitly characterized in a data-driven manner. The identification of this resilient cluster underscores the potential for protective psychological or behavioral factors in modulating fatigue burden and highlights the importance of personalized supportive interventions [53,54,55,56,57].

The three identified clusters provide meaningful insights into the heterogeneity of fatigue experiences among patients with MFS. Cluster 1 was characterized by younger age, lower PHQ-9 and ISI scores, and a lower BMI. These features suggest the presence of potential protective factors, such as fewer comorbidities, higher physiological reserve, and possibly greater psychological resilience [58]. This subgroup might benefit from standard monitoring, focusing on maintaining physical and psychological well-being to preserve their favorable fatigue status. Cluster 2 showed intermediate levels of fatigue severity alongside higher rates of depressive symptoms and subthreshold insomnia. This indicates a subgroup where psychosocial vulnerability plays a prominent role, despite relatively lower physical burden [59]. In this profile, the early identification of emotional distress and tailored psychosocial support, including psychological counseling and sleep interventions, could help prevent progression to higher fatigue states. Cluster 3 included older patients with higher BMI, longer disease duration, and the highest levels of depressive symptoms and insomnia severity. This profile reflects the cumulative burden of physical comorbidities and psychological distress [60]. Patients in this group are likely to require more intensive, multidisciplinary management strategies that address both somatic and mental health dimensions of fatigue.

The identification of distinct fatigue profiles among individuals with MFS has important clinical and practical implications. Profiling patients into homogeneous subgroups allows clinicians to move beyond a “one-size-fits-all” approach and tailor interventions to the specific needs of each profile [61]. For example, patients in the “low fatigue and minimal psychological and sleep-related symptoms” group (Cluster 1) may primarily benefit from regular monitoring and general lifestyle counseling to maintain their favorable status. In contrast, those in the “moderate fatigue with moderate psychological and sleep-related symptoms” group (Cluster 2) could require targeted psychological support, such as counseling, cognitive-behavioral therapy, or sleep hygiene interventions, to address emotional distress and subthreshold insomnia that may exacerbate fatigue [62]. Finally, the “high fatigue with elevated psychological and sleep-related symptoms” (Cluster 3) highlights a subgroup in need of intensive, multidisciplinary care strategies that simultaneously address comorbid physical conditions, psychological distress, and severe fatigue [63].

In addition to the cluster-based characterization, exploratory analyses were conducted to clarify further the relationships among age, psychosocial burden, and fatigue severity. Spearman correlations showed that fatigue severity was most strongly related to depressive symptoms (ρ = 0.62, *p* < 0.001) and, to a lesser extent, insomnia severity (ρ = 0.41, *p* < 0.001), whereas its association with age was weaker (ρ = 0.22, *p* = 0.014). Partial correlations controlling for age confirmed that both depressive symptoms (ρ = 0.60, *p* < 0.001) and insomnia severity (ρ = 0.39, *p* < 0.001) remained significantly associated with fatigue, indicating that these relationships are largely independent of chronological age. To examine variability within older participants, those in the highest age tertile (≥47 years) were stratified by fatigue status using the clinical FSS cut-off. High-fatigue older adults reported substantially higher depressive symptom scores and a trend toward higher insomnia scores, while the BMI and disease duration did not differ. These findings suggest that even among older individuals, lower levels of depressive symptoms and potentially better sleep quality may serve as protective factors against severe fatigue. This underscores the importance of addressing modifiable psychosocial factors in targeted intervention strategies, especially for high-risk subgroups, even though these hypotheses from explorative analyses require further empirical testing in future studies.

This study has several limitations that should be acknowledged. First, its cross-sectional design precludes establishing causal relationships between fatigue and the associated clinical or psychosocial factors identified across clusters. Longitudinal studies are necessary to confirm whether these profiles remain stable over time and to investigate the trajectories of fatigue severity and its associated factors. Second, the reliance on self-reported measures for key variables such as fatigue severity (FSS), depressive symptoms (PHQ-9), and insomnia (ISI) introduces potential reporting and recall biases. Although these instruments are validated and widely used, self-report data may be influenced by individual perceptions or social desirability. Third, while our sample was carefully selected and achieved a high response rate, it consisted of patients from a single specialized center in Italy, which may limit the generalizability of findings to other populations or healthcare settings. Additionally, there is a potential for selection bias as patients who agreed to participate may be more motivated or health-conscious, potentially leading to an underestimation of fatigue severity in the broader MFS population. Future studies incorporating objective measurements (e.g., actigraphy for fatigue, clinical interviews for psychiatric symptoms), broader sampling, and longitudinal follow-up would help validate and expand on these findings.

## 5. Conclusions

This study identified and characterized three distinct fatigue-related profiles in individuals with MFS using an integrative, data-driven cluster analysis approach. The combination of t-SNE dimensionality reduction and hierarchical clustering allowed for the identification of clinically meaningful subgroups that differ in terms of age, psychosocial vulnerability, and fatigue severity. These findings highlight the heterogeneity of fatigue experiences in this population and underscore the importance of personalized assessment and management strategies tailored to individual needs. Future research should confirm these profiles longitudinally and explore tailored interventions to improve quality of life and clinical outcomes in patients with MFS.

## Figures and Tables

**Figure 1 jcm-14-05802-f001:**
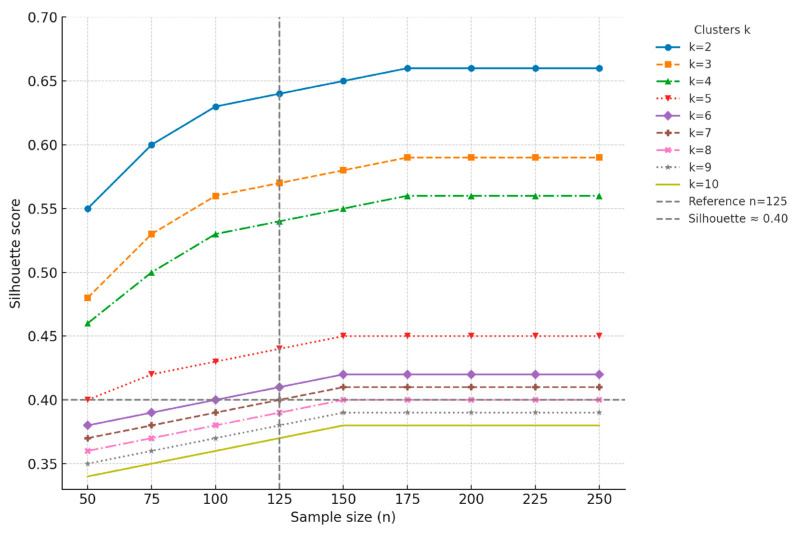
Simulated relationship between sample size and silhouette score across different cluster solutions (k = 2 to 10).

**Figure 2 jcm-14-05802-f002:**
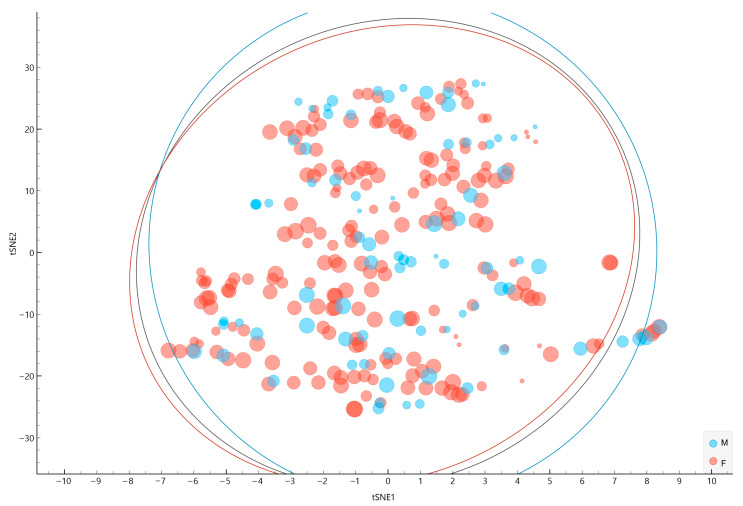
The two-dimensional t-SNE projection of the sample colored by sex and sized by fatigue severity. Legend: Colors—blue = male, red = female. Point size—proportional to Fatigue Severity Scale (FSS) score. Ellipses—contain the approximate distribution of each sex group (the gray ellipse is the aggregated information).

**Figure 3 jcm-14-05802-f003:**
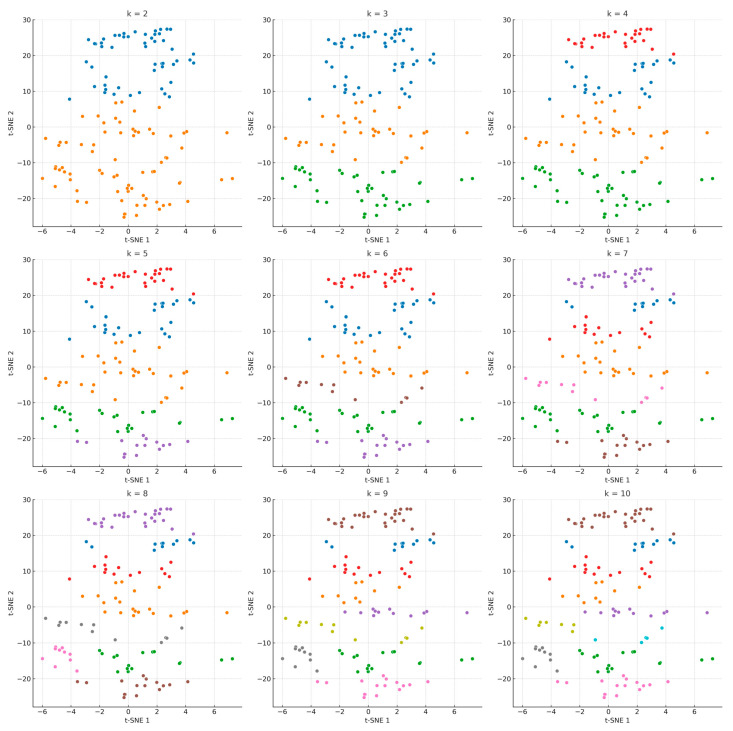
t-SNE scatterplots displaying cluster solutions from k = 2 to k = 10. Note. Each plot shows participant distribution in the t-SNE space, colored by cluster membership.

**Figure 4 jcm-14-05802-f004:**
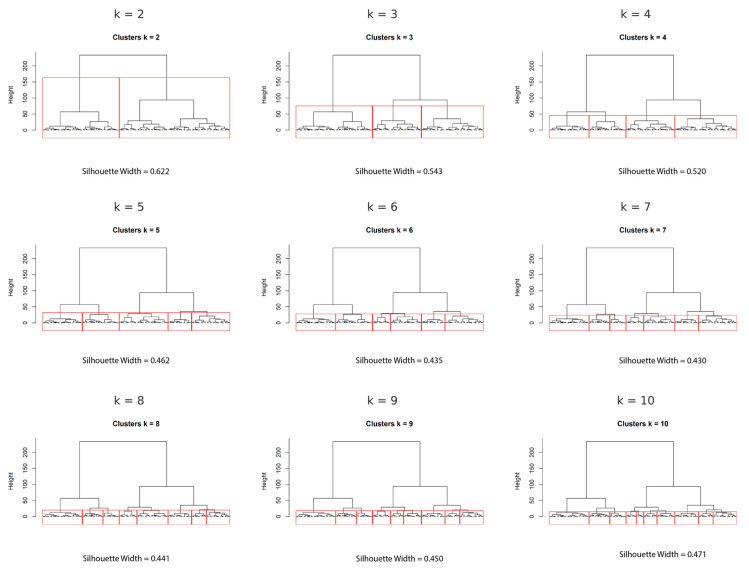
Dendrograms from hierarchical clustering for k = 2 to k = 10.

**Figure 5 jcm-14-05802-f005:**
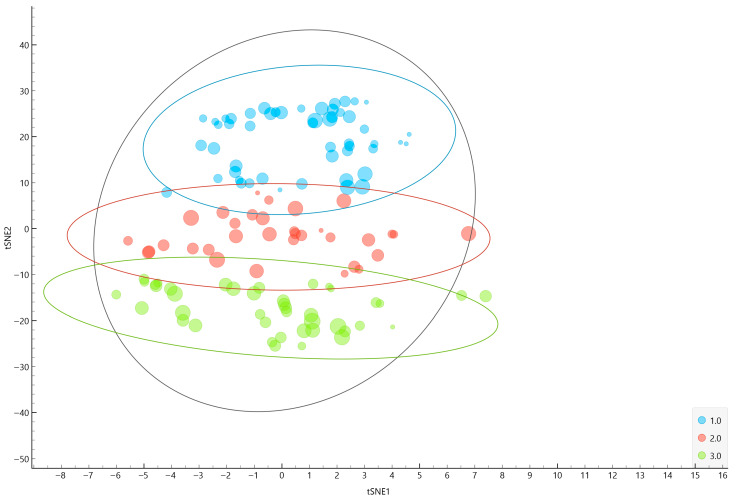
A t-SNE scatter plot colored by cluster membership, sized by fatigue severity. Legend: Points are colored according to final cluster assignment (Cluster 1 = low-fatigue resilient profile, Cluster 2 = moderate-fatigue psychosocially vulnerable profile, Cluster 3 = high-fatigue burdened profile) and scaled by individual FSS scores to reflect fatigue severity. Ellipses represent the estimated 95% confidence boundaries of each cluster in the t-SNE space, illustrating the separation and cohesion of the identified profiles.

**Table 1 jcm-14-05802-t001:** Summary of sample characteristics.

Characteristic		N (%)
Sex		
	Male	61 (48%)
	Female	66 (52%)
Age		
	Years (mean ± SD)	38.75 ± 14.06
	Primary school	2 (1.6%)
	Lower secondary school	30 (23.6%)
	Higher secondary school	48 (37.8%)
	University	47 (37.0%)
	Years since diagnosis, median (IQR)	13 (7–23)
Profession		
	Active worker—office	71 (55.9%)
	Active worker—home	8 (6.3%)
	Occasional worker	3 (2.4%)
	Retired	3 (2.4%)
	Unemployed	42 (33.1%)
Cardiovascular comorbidities		102 (80.3%)
	Hypertension	11 (8.7%)
	Cardiovascular medications	97 (76.4%)
	Respiratory diseases	18 (14.2%)
Other diseases		104 (81.9%)
	Visual impairments	56 (44.1%)
	Thyroid dysfunction	12 (11.5%)
	Neuropathies	3 (2.9%)
	Joint diseases	8 (7.7%)
	Scoliosis	71 (68.3%)
	Multiple conditions	1 (0.96%)
BMI		
	kg/m^2^ (median; IQR)	21.33 (19.11–23.41)
PHQ-9		
	Score (mean ± SD)	6.01 ± 4.51
	No depressive symptoms	52 (40.9%)
	Minimal depressive symptoms	54 (42.5%)
	Minor depression	15 (11.8%)
	Moderate major depression	4 (3.1%)
	Severe major depression	2 (1.6%)
ISI		
	Score (mean ± SD)	7.02 ± 4.13
	No clinically significant insomnia	78 (61.4%)
	Subthreshold insomnia	43 (33.9%)
	Clinical insomnia (moderate)	6 (4.7%)
FSS		
	Score (mean ± SD)	3.88 ± 1.68
	Clinically relevant fatigue	40 (31.5%)
Fatigue in the last two weeks		
	Never	9 (7.6%)
	Sometimes	80 (67.2%)
	Every day	30 (25.2%)
Fatigue pattern		
	In the morning	25 (21.0%)
	In the afternoon	78 (65.5%)
	All day	16 (13.4%)

Legend: Summary of sociodemographic, clinical, and psychosocial characteristics of the included patients with Marfan syndrome (N = 127). Values are presented as numbers and percentages for categorical variables and as mean ± standard deviation (SD) or median with interquartile range (IQR) for continuous variables, as appropriate. PHQ-9 = Patient Health Questionnaire-9; scores range from 0 to 27, with higher scores indicating more severe depressive symptoms. ISI = Insomnia Severity Index; scores range from 0 to 28, with higher scores indicating more severe insomnia. FSS = Fatigue Severity Scale; scores range from 1 to 7, with higher scores indicating greater fatigue severity.

**Table 2 jcm-14-05802-t002:** Clinical, demographic, and psychosocial characteristics across the three identified clusters (k = 3).

Characteristic	Cluster 1 (N = 49)	Cluster 2 (N = 32)	Cluster 3 (N = 46)	*p*-Value
Age (mean ± SD)	22.6 ± 3.1	36.4 ± 5.9	53.5 ± 6.1	<0.001
Years since diagnosis (median, IQR)	8.6 (6.6–10.0)	12.1 (9.7–15.0)	15.6 (13.7–19.0)	0.0027
BMI (median, IQR)	20.0 (18.4–21.4)	21.8 (19.5–23.6)	23.9 (22.5–25.6)	<0.001
Sex: male	20 (40.8)	17 (53.1)	24 (52.2)	0.137
Sex: female	14 (59.2)	23 (46.9)	29 (47.8)	
PHQ-9 (mean ± SD)	3.5 ± 2.6	5.8 ± 3.7	7.2 ± 3.6	0.016
No depressive symptoms	18 (36.7)	15 (46.9)	19 (41.3)	
Minimal depressive symptoms	12 (24.5)	16 (50.0)	26 (56.5)	
Minor depression	4 (8.2)	5 (15.6)	6 (13.0)	
Moderate major depression	0	3 (9.4)	1 (2.2)	
Severe major depression	0	1 (3.1)	1 (2.2)	
ISI (mean ± SD)	4.3 ± 3.7	6.7 ± 4.5	8.8 ± 4.8	0.029
No insomnia	22 (44.9)	23 (71.9)	33 (71.79)	
Subthreshold insomnia	11 (22.4)	16 (50.0)	17 (37.09)	
Clinical insomnia (moderate)	1 (2.0)	1 (3.1)	3 (6.5)	
FSS (mean ± SD)	2.7 ± 0.8	3.5 ± 1.1	4.9 ± 1.0	<0.001
Clinically relevant fatigue	0	9 (25.1)	31 (67.4)	<0.001

Legend: BMI, body mass index; IQR, interquartile range; SD, standard deviation; PHQ-9, Patient Health Questionnaire-9; ISI, Insomnia Severity Index; FSS, Fatigue Severity Scale.

## Data Availability

Data are available upon reasonable request. The datasets generated and/or analyzed during the current study are not publicly available due to privacy or ethical restrictions but are available from the corresponding author upon reasonable request.

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
