# Peer review of "Phenotyping Fatigue Profiles in Marfan Syndrome Through Cluster Analysis: A Cross-Sectional Study of Psychosocial and Clinical Correlates"

_jcm, 2025, doi:10.3390/jcm14165802_

Round 1

Reviewer 1 Report

Comments and Suggestions for Authors

The presented manuscript by N.T. Udugampolage and co-authors aims to assess fatigue symptom as a psychosocial and clinical problem in Marfan Syndrome employing cluster analysis method.

The topic is interesting and resonable, and the data presented in the text is actual and important in medical practice. I find this work wery interesting and well done: the authors analysed various parameters of fatigue criteria and provide inportant results.

I would like to mention the good statistical analysis that supports author's claims as well as inclusion/exclusion criteria that are given correctly.

The literature discussed is well-structured and corresponds to the topic. Additionally, almost all the sources cited are published in last 5 years, a lot of references are published in 2024-2025.

The findings of the authors will be useful for a diverse readership and medical practitioners.

No remarks or corrections are adressed to the authors from my side, I can only congratulate the authors with the nice job done.

Author Response

COMMENT 1: The presented manuscript by N.T. Udugampolage and co-authors aims to assess fatigue symptom as a psychosocial and clinical problem in Marfan Syndrome employing cluster analysis method.

The topic is interesting and resonable, and the data presented in the text is actual and important in medical practice. I find this work wery interesting and well done: the authors analysed various parameters of fatigue criteria and provide inportant results.

I would like to mention the good statistical analysis that supports author's claims as well as inclusion/exclusion criteria that are given correctly.

The literature discussed is well-structured and corresponds to the topic. Additionally, almost all the sources cited are published in last 5 years, a lot of references are published in 2024-2025.

The findings of the authors will be useful for a diverse readership and medical practitioners.

No remarks or corrections are adressed to the authors from my side, I can only congratulate the authors with the nice job done.

RESPOND: Thank you for your revision and the appreciation of our work.

Reviewer 2 Report

Comments and Suggestions for Authors

Dear authors, thank you for the opportunity to review the results of your study.
The study of fatigue profiles in Marfan syndrome is important and relevant, as it allows for the development of more targeted strategies for supporting such people. There is a lack of scientific knowledge in the classification of fatigue manifestations and its correlates in people with Marfan syndrome.
The strengths of the study are:
- well-developed study of the issue of studying fatigue and related factors in people with Marfan syndrome;
- clear justification and representativeness of the sample size;
- detailed and thorough description of the design and stages of the study, statistical methods and all assumptions, the use of reliable questionnaires to assess fatigue, depressive symptoms and insomnia;
- high clarity in the presentation of results (a large number of figures and 2 tables);
- high-quality discussion of the results, description of the limitations of the study and future directions;
- clear wording in the conclusions.
At the same time, there are some recommendations and questions for the authors that require clarification:
1. The purpose and hypothesis of the study should be specified more precisely.
2. The authors obtained three clusters that are clearly differentiated by the levels of fatigue (low, medium and high), age (from younger to older) and the corresponding symptoms of Marfan syndrome. When interpreting the results, did the authors assume a linear relationship between these characteristics: the older the person, the more pronounced the symptoms of Marfan syndrome, the more pronounced the fatigue? Reading these data suggests this idea, but it is not discussed in any way. The authors could test these assumptions on their sample, add data on these results. And also clarify which correlates could provide greater variability? For example, if we take groups with an older age and select those with more and less pronounced symptoms, and compare the manifestations of fatigue and associated factors. This would allow us to identify factors that reduce adverse effects and promote well-being.
The comments presented do not detract from the overall positive impression of the study.

Best wishes, reviewer

Author Response

Comment 1: Dear authors, thank you for the opportunity to review the results of your study.
The study of fatigue profiles in Marfan syndrome is important and relevant, as it allows for the development of more targeted strategies for supporting such people. There is a lack of scientific knowledge in the classification of fatigue manifestations and its correlates in people with Marfan syndrome.
The strengths of the study are:
- well-developed study of the issue of studying fatigue and related factors in people with Marfan syndrome;
- clear justification and representativeness of the sample size;
- detailed and thorough description of the design and stages of the study, statistical methods and all assumptions, the use of reliable questionnaires to assess fatigue, depressive symptoms and insomnia;
- high clarity in the presentation of results (a large number of figures and 2 tables);
- high-quality discussion of the results, description of the limitations of the study and future directions;
- clear wording in the conclusions.

Response 1: Thank you for your review and the appreciation of the strengths of our study.

Comment 2: At the same time, there are some recommendations and questions for the authors that require clarification:
1. The purpose and hypothesis of the study should be specified more precisely.

Response 2: We thank the reviewer for this valuable suggestion. In the revised manuscript, we have refined the final paragraph of the Introduction to provide a more precise statement of purpose and to explicitly include our study hypotheses.

Comment 3: 2. The authors obtained three clusters that are clearly differentiated by the levels of fatigue (low, medium and high), age (from younger to older) and the corresponding symptoms of Marfan syndrome. When interpreting the results, did the authors assume a linear relationship between these characteristics: the older the person, the more pronounced the symptoms of Marfan syndrome, the more pronounced the fatigue? Reading these data suggests this idea, but it is not discussed in any way. The authors could test these assumptions on their sample, add data on these results. And also clarify which correlates could provide greater variability? For example, if we take groups with an older age and select those with more and less pronounced symptoms, and compare the manifestations of fatigue and associated factors. This would allow us to identify factors that reduce adverse effects and promote well-being.

Response 3: We thank the reviewer for this insightful suggestion. To address it, we performed additional exploratory analyses to examine potential monotonic relationships between age, psychosocial burden, and fatigue severity. Specifically, we computed Spearman correlations and partial correlations (controlling for age) among fatigue severity, depressive symptoms, insomnia severity, and clinical variables. Furthermore, we compared low- versus high-fatigue subgroups within the highest age tertile (≥47 years) to identify factors potentially moderating fatigue burden in older participants. These analyses have been added to the Methods (data analysis), Results (new subsection “Exploratory associations between age, psychosocial factors, and fatigue”), and Discussion. The findings indicate that depressive symptoms, and to a lesser extent, insomnia, are key correlates of fatigue independent of age, and may act as protective or aggravating factors in older individuals with comparable disease duration and BMI.

Comment 4: The comments presented do not detract from the overall positive impression of the study.

Response 4: Thank you for the appreciation of our work.